# Impact of Arbuscular Mycorrhizal Fungi, Phosphate Solubilizing Bacteria and Selected Chemical Phosphorus Fertilizers on Growth and Productivity of Rice

Nehal M. Elekhtyar [1], Mamdouh M. A. Awad-Allah [1,*], Khalid S. Alshallash [2], Aishah Alatawi [3], Rana M. Alshegaihi [4] and Reem A. Alsalmi [5]

[1] Rice Research Department, Field Crops Research Institute, Agricultural Research Center, Kafr El-Sheikh 33717, Egypt
[2] College of Science and Humanities, Huraymila, Imam Mohammed Bin Saud Islamic University (IMSIU), Riyadh 11432, Saudi Arabia
[3] Biology Department, Faculty of Science, University of Tabuk, Tabuk 71421, Saudi Arabia
[4] Department of Biology, College of Science, University of Jeddah, Jeddah 21493, Saudi Arabia
[5] Department of Biology, College of Science, AlBaha University, Al Baha 1988, Saudi Arabia
* Correspondence: mamdouhrice@gmail.com

**Abstract:** Phosphorus is the second most significant macro nutrient in rice productivity. Phosphorus fixation in Egyptian soil makes it unavailable for rice to absorb. The goal of this study was to examine the effects of microbial and chemical sources of phosphorus fertilizers on the Egyptian Sakha 106 rice cultivar by applying different sources of phosphorus to increase the bioavailability of soil phosphorus for plants and to allow it to be fixed biologically to change it from an insoluble form to a soluble and available form for rice to absorb. So, in the 2019 and 2020 seasons, a field experiment was conducted at the experimental farm of Sakha Agricultural Research Station, Sakha, Kafr El-Sheikh, Egypt. The experiment was carried out using a Randomized Complete Block Design with four replications to determine the best phosphorus source for rice and soil among various treatments, which included 100% single super phosphate (SSP) basal application (P1), 75% single super phosphate (SSP) basal application (P2), P2 + phosphate-solubilizing bacteria (PSBs) top-dressing, P2 + arbuscular mycorrhizal fungi (AMFs) top-dressing P2 + phosphorus nanoparticles (PNPs) foliar spraying, P2 + phosphoric acid (PA) foliar spraying, P2 + (PSBs + AMFs) foliar spraying, P2 + (PSBs + PNPs) foliar spraying, P2 + (PSBs + PA) foliar spraying, P2 + (PNPs + PA) foliar spraying, P2 + (PSBs + PNPs + PA) foliar spraying and zero-phosphorus fertilizer. The results showed that the highest values were mostly obtained using the combination of 75% SSP basal application with the foliar spraying of PSBs, PNPs and PA, with substantial beneficial impacts on the leaf area index (3.706 and 3.527), dry matter accumulation (464.3 and 462.8 g m$^2$), plant height (96.33 and 95.00 cm), phosphorus uptake in grain (24.3 and 24.49 Kg ha$^{-1}$), phosphorus uptake in straw (17.7 and 17.0 Kg ha$^{-1}$) and available phosphorus in the soil at harvest (21.75 and 21.70 ppm) in the 2019 and 2020 seasons, respectively; moreover, 75% SSP basal application with the foliar spraying of PSBs, PNPs and PA or 100% SSP basal application alone improved the number of panicles (506.3 or 521.1 and 521.9 or 547.1 m$^{-2}$), filled grain weight (3.549 or 3.534 and 3.627 or 3.767 g panicle$^{-1}$), the percentage of filled grain (96.19 or 96.47 and 95.43 or 96.24%), grain yield (9.353 or 9.221 and 9.311 or 9.148 t ha$^{-1}$) and straw yield (11.51 or 11.46 and 11.82 or 11.69 t ha$^{-1}$) in the 2019 and 2020 seasons, respectively. Chemical P fertilizers combined with the foliar spraying of PSBs, PNPs and PA obtained the highest crop productivity and improved most of the examined characteristics without any significant changes with respect to chemical P application alone in some other characteristics, followed by 75% SSP + top-dressing with PSBs + AMFs. The treatment that included the combination of 75% SSP basal application and the foliar spraying of PSBs +PNPs +PA is recommended, as it might be utilized to boost rice yield by solubilizing P in soil and increasing the absorption efficiency. In addition, it reduces chemical P fertilizers by 25%, which would guarantee a cleaner environment and soil conservation.

**Keywords:** rice; single super phosphate; phosphate-solubilizing bacteria; arbuscular mycorrhizal fungi; phosphorus nanoparticles; phosphoric acid

## 1. Introduction

Rice is widely cultivated in Egypt, with an annual cultivated area of about 1,307,000 hectares producing 5.27 million tons, with an average of 9.59 tons per hectare [1]. Nitrogen, phosphorus and potassium are the vital plant nutrients for rice. Phosphorus (P) is the second prime macronutrient after nitrogen for rice productivity [2]. Plants need to absorb P from the soil. P is present in three forms in soil, i.e., soluble inorganic P, insoluble inorganic P and organic P [3]. Phosphorus has a vital role in plant metabolic processes, such as cell division, photosynthesis, nutrient transfer into the plant and the relocation of genetic traits [4]. P is entangled in many plant metabolic tasks, and it has a main role in the organizing of enzymes [5,6]. P is a vital part of adenosine diphosphate and triphosphate organic molecules, which are applied for the transfer and storage of cell energy. It is an important component of ATP, linked to cell energy [7], and a needed component of DNA and RNA for plant structure [8]. P is important for the growth, seedling vigor, healthy roots, crop quality and yield of the plant. Phosphorus fixing in Egyptian soil is a big problem in agricultural production. P cannot move far in the soil to go to the roots [9]. The salts of phytic acid are the main form of organic P in the soil and are not available to plants; it is fixed in soil colloids in Ca, Al and Fe phosphate forms [10]. The efficiency of inorganic P fertilizers is very low, at only 10 to 20% [11], and phosphate is only absorbed by plants as monobasic $(H_2PO_4)^-$ and dibasic $(HPO_4)^{-2}$ ions. A large amount of soluble inorganic phosphate applied to soil via inorganic fertilizers is rapidly mobilized and unavailable to plants shortly after application. Thus, increasing soil phosphorus availability requires the release of insoluble and fixed forms of phosphorus [12]. Some soil microorganisms can mineralize and solubilize P from organic and inorganic forms. Soil microorganisms, primarily those of the genera phosphate-solubilizing bacteria and arbuscular mycorrhizal fungi, can convert insoluble phosphates to soluble forms by secreting organic acids [13]. On the other hand, conventional farming heavily relies on the application of chemical phosphorous fertilizers to maintain optimal levels of phosphorous in agricultural soils. The majority of phosphates in the soil are absorbed by soil particles or incorporated into soil organic matter [14]. A large portion of the soluble phosphate applied via soil chemical fertilizers is quickly immobilized and rendered inaccessible to plants. Thus, P is lost via leaching losses, causing environmental pollution problems. Current studies attempt to overcome this condition by investigating alternative sources that are both cost effective and environmentally friendly [15]. Superphosphate has enough amount of calcium residues, and 50% of this phosphorus is in the form of dicalcium phosphate [16,17]. A viable strategy for increasing rice productivity is to use rhizosphere microbial modification to increase yield and the availability of minerals. Microorganisms play a vital role in rice by storing nutrients in plants and minimizing the demand for inorganic fertilizers [13,15]. Microorganisms help plants by solubilizing fixed P in soil and boosting P uptake efficiency. Microorganisms are very active in the rhizosphere zone in soil [18] and are essential for enzyme activity, i.e., invertase, catalase, urease, neutral phosphatase and alkaline phosphatase, in the rhizosphere, which has a good effect on the concentrations and mineralization of available P in soil [19,20]. Soluble P is released from insoluble P by types of solubilization microorganisms such as arbuscular mycorrhizal fungi (AMFs) [21] and phosphate-solubilizing bacteria (PSBs) [22]. PSBs and AMFs can solubilize P in soil and decrease the input of mineral fertilizers [5,6,23]. Biofertilizers are a substitutional source of inorganic fertilizers [24]. Arbuscular mycorrhizal fungi (AMFs) facilitate the ability of plants to absorb water and nutrients from the soil interphase. However, the fungus uses the carbon from the plant to fuel its development, advancement and other physiological functions [25]. AMFs colonize roots in the rhizosphere zone and improve phosphorus absorption from the insoluble form

to the soluble form [26]. AMFs can boost P uptake via the mycorrhizal root system [27]. Under conditions of water deficit, the symbiotic AMF inoculation of plant roots can be delivered and maintained, increasing plant water acquisition and boosting plant growth and crop output [28]. Phosphate-solubilizing bacteria (PSBs) have senior value for plant nutrition and perform a major role in plant-growth-promoting rhizobacteria (PGPRs) as biofertilizers of crops [22,29]. PSBs have the ability to solubilize insoluble mineral phosphate compounds, such as tricalcium phosphate, dicalcium phosphate, hydroxyapatite and rock phosphate [30]. PSBs have the ability to dissolve P ions attached to Al, Fe, Ca and Mg soil cations and then transform them into forms that plants can naturally absorb [13]. PSBs can change insoluble phosphate by secreting organic acids, such succinic, fumaric, acetic, formic and propionic acids. Plant growth can be increased through rice root colonization by PSBs, and the application of efficient PSBs such as Bacillus megaterium can increase soil P availability by nearly 30% [31]. The important aspect of a successful PSB–root interaction is the best fit in terms of rooting stimulation in addition to rhizosphere P solubilization [32]. Nanotechnology is very favorable in the field of agriculture [33]. Nanoparticles have the ability to increase plant metabolism [34]. Moreover, when compared with chemical fertilizers' requirements and costs, nanofertilizers are cheaper and are required in lower amounts. Using nanoparticles for the growth of plants and for the control of plant diseases is a recent practice. Hence, it is crucial to optimize the use of chemical fertilization to fulfill crop nutrient requirements and minimize the risk of environmental pollution. Nanoparticles have unique physicochemical properties and the potential to boost plant metabolism [34]. Nanoparticles were recently used to improve the growth of plants [35–39]. Rock phosphate is used in the form of a nanoparticle to raise the availability of P to crop plants, and it prevents P fixation in the soil [40,41]. The purpose of this study was to determine the effect of microbial and chemical phosphorous fertilizers on Egyptian rice variety Sakha 106 through the use of different sources of phosphorous. In addition, we studied the extent to which phosphorous can be biologically repaired by changing it from an insoluble form to a soluble and available form for uptake by rice in the rhizosphere. On the other hand, we investigated the definition and application of the best combination of biofertilizers and part of P chemical fertilizers to reduce the used amounts of chemical fertilizers in order to reduce the leaching of phosphorous from rice soil and reduce pollution.

## 2. Materials and Methods

### 2.1. Field Experiment

A field experiment was conducted in two seasons, 2019 and 2020, at the experimental farm of Sakha Agricultural Research Station, Sakha, Kafr El-Sheikh, Egypt (30°57′12″ north latitude, 31°07′19″ east longitude), to evaluate the effect of phosphate-solubilizing bacteria, arbuscular mycorrhizal fungi and selected chemical phosphorus fertilizers on the growth and productivity of rice. Representative soil samples were taken from each site at the depth of 0–30 cm from the soil surface. Samples were air-dried, ground to pass through a 2 mm sieve and mixed well. The procedures of the soil analyses followed the methods by [42]. The results of the chemical analyses in both seasons at the experimental site are shown in Table 1.

A variety, Sakha 106, with high yielding potential was used. To improve early germination, seeds were soaked in water for 24 h and then incubated for 48 h at a rate of 96 kg ha$^{-1}$. On 10th of May in both seasons, pre-germinated seeds were uniformly broadcasted in the nursery. The permanent field was prepared by plowing and then wet-leveling. Rice seedlings were carefully removed from the nursery and put into plots in the field 30 days after sowing; they were then manually transplanted into 12 m$^2$ subplots in $20 \times 20$ cm$^2$ spaces between rows and hills at three seedlings hill$^{-1}$. From transplanting to two weeks before harvesting, plots were kept flooded. Water was removed from the plots two weeks before harvesting. The preceding crop was barley.

**Table 1.** Soil chemical properties of the experimental sites in the 2019 and 2020 seasons.

| | 2019 | 2020 |
|---|---|---|
| Soluble anions (meq. $L^{-1}$) | | |
| $HCO_3^-$ | 17.80 | 17.00 |
| $Cl^-$ | 17.20 | 16.90 |
| $SO_4^{--}$ | 3.12 | 2.90 |
| Soluble Cations (meq. $L^{-1}$) | | |
| $Ca^{++}$ | 9.41 | 8.25 |
| $Mg^{++}$ | 4.52 | 3.80 |
| $K^+$ | 1.48 | 1.22 |
| $Na^{++}$ | 12.40 | 13.05 |
| Available micronutrients (ppm) | | |
| $Fe^{++}$ | 5.95 | 5.30 |
| $Mn^{++}$ | 3.30 | 3.10 |
| $Zn^{++}$ | 1.00 | 1.15 |
| Available $NH_{4+}$ (mg $kg^{-1}$) | 14.15 | 13.70 |
| Available P (mg $kg^{-1}$) | 11.92 | 12.00 |
| Available K (mg $kg^{-1}$) | 375 | 380 |
| Ec (ds.$m^{-1}$) | 2.55 | 2.25 |
| pH (1:2.5 water suspension) | 8.12 | 8.18 |
| Organic matter (O.M) % | 1.59 | 1.53 |
| Soil texture | Clayey | Clayey |

*2.2. Chemical Fertilizers*

1. Nitrogen fertilizer was applied at the rate of 165 kg N $ha^{-1}$ in the form of urea (46.5% N). Urea was applied in two doses to each plot, with the first $^2/_3$ of dosage being employed as basal application. A total of 30 days after transplanting (DAT), the other $^1/_3$ of dosage was used for top-dressing;

2. A chemical phosphorus fertilizer in the form of single super phosphate (SSP) with 15.5% $P_2O_5$ was added and incorporated well into the soil at the time of final land preparation as basal application at the rate of 36 kg $ha^{-1}$;

3. Orthophosphoric acid ($H_3PO_4$) with 85% phosphoric acid was used. In the booting stage (25 days after transplanting), a liquid solution containing 120 mg $L^{-1}$ per ha was used for foliar spraying;

4. Hydroxyapatite ($Ca_5 (PO_4)_3 OH$) nanoparticles were used as phosphorus nanoparticles. In the booting stage, a liquid solution of 2400 mg $L^{-1}$ per ha was sprayed on the leaves as foliar spray. Transmission electron microscopy (TEM) was performed to characterize the size distribution and shape of the synthesized phosphorus nanoparticles (Figure 1) [43].

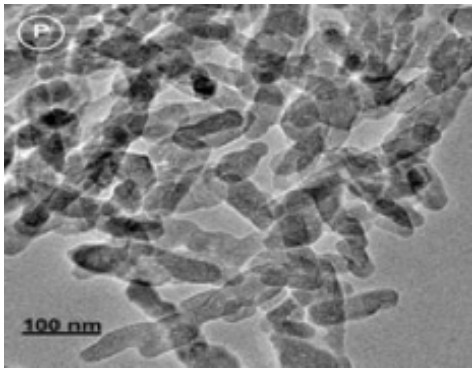

**Figure 1.** Transmission electron microscopy (TEM) of phosphorus nanoparticles.

### 2.3. Biological Fertilizers

Two biological phosphorus sources were used, i.e.:

1.  Phosphate-solubilizing bacteria (PSBs), which included Bacillus megatherium. PSBs were utilized with two different techniques. In the first, at a rate of 2.380 kg ha$^{-1}$ via powder inoculation, PSBs were used for top-dressing during transplanting in the permanent field 30 days after sowing. Inoculation powder was combined with enough sand to make homogeneous dispersion easier. The other method involved spraying a foliar solution at a rate of 2.975 L ha$^{-1}$ in the booting stage (25 days after transplanting).
2.  Arbuscular mycorrhizal fungi (AMFs), which included Glomus sp. AMFs were employed at a rate of 7.200 K ha$^{-1}$ via powder inoculation and were used for top-dressing upon transplanting in the permanent field 30 days after sowing. Inoculation powder was combined with enough sand to make homogeneous dispersion easier.

The biofertilizers were obtained from Agric. Microbial. Dept., Soil and Water Institute, Agricultural Research Center, Giza, Egypt's Ministry of Agriculture and Land Reclamation's General Organization for Agricultural Equalization Fund. Rice Research and Training Center recommended that the traditional agricultural practices of cultivating rice be followed by [4].

### 2.4. Experiment Treatments

The experiments were carried out using a Randomized Complete Block Design with four replications in two seasons, 2019 and 2020, to determine the best P source for rice and soil among various treatments, which included: T1—basal application of 100% single super phosphate (SSP) ($P_1$); T2—basal application of 75% SSP ($P_2$); T3—$P_2$ + top-dressing with phosphate-solubilizing bacteria (PSBs); T4—$P_2$ + top-dressing with arbuscular mycorrhizal fungi (AMFs); T5—$P_2$ + foliar spraying of phosphorus nanoparticles (PNPs); T6—$P_2$ + foliar spraying of phosphoric acid (PA); T7—$P_2$ + top-dressing with PSBs + AMFs; T8—$P_2$ + foliar spraying of PSBs + PNPs; T9—$P_2$ + foliar spraying of PSBs + PA; T10—$P_2$ + foliar spraying of PNPs + PA; T11—$P_2$ + foliar spraying of PSBs + PNPs + PA; T12—zero-phosphorus fertilizer (Control).

### 2.5. Studied Characteristics

Five hills from each plot were randomly collected to estimate the leaf area index (LAI) in the booting stage, dry matter accumulation (g m$^{-2}$) in the booting stage, plant height (cm) at harvest and number of panicles m$^{-2}$ at harvest. At harvest, the panicles of ten hills for each plot were taken to determine filled grain weight (g) panicle$^{-1}$ and the percentage of filled grain (%). The plants of six inner rows of each plot were harvested separately at full maturity, dried and threshed; then, the grain yield was adjusted to 14% moisture content. Grain and straw yields were recorded and converted into t ha$^{-1}$. Samples of grain were taken at harvest to estimate phosphorus uptake in grain and straw (Kg ha$^{-1}$). Phosphorus percentages in dry grain and straw samples at harvest were calculated as described by [44] and measured using a spectrophotometer using the ascorbic acid method [45]; then, phosphorus uptake was determined as follows: P% × grain or straw dry weight. Soil samples were taken at harvest from each plot at the depth of 0–30 cm from the soil surface. Samples were air-dried, ground to pass through a 2 mm sieve and mixed well to estimate the available phosphorus in soil (ppm) [44]. The LAI was calculated according to Watson [46]:

$$LAI = \frac{\text{Leaf area of fixed number of hills}}{\text{Ground area occupied by these hills m}^2}$$

### 2.6. Statistical Analyses

The data were statistically analyzed using the technique of the analysis of variance (ANOVA) of the Randomized Complete Block Design (RCBD) in four replicates as described

by [47]. The treatment means were compared using Duncan's Multiple Range Test as described by [48]. All statistical analyses were performed using the "MSTAT-C" computer software package [49,50].

## 3. Results

### 3.1. Growth Characteristics

Table 2 shows that the combination of 75% single super phosphate (SSP) basal application and the foliar spraying of phosphate-solubilizing bacteria (PSBs) + phosphorus nanoparticles (PNPs) + phosphoric acid (PA) significantly increased the leaf area index (LAI) in the booting stage (3.706 and 3.527) in 2019 and 2020, respectively, with no significant differences when compared with 100% SSP, followed by 75% SSP and top-dressing with PSBs + AMFs or 75% SSP + PSBs + PNPs. In the two seasons of 2019 and 2020, low values were recorded for the zero-P fertilizer (1.960 and 1.667), respectively. Dry matter accumulation in g m$^2$ (DMA) in the booting stage significantly increased following the application of the combination of 75% SSP basal application and the foliar spraying of PSBs + PNPs + PA (464.3 and 462.8 g m$^2$) in 2019 and 2020, respectively, followed by 100% SSP, while low values were recorded for the zero-P fertilizer (179.3 and 194.6 g m$^2$) in the two seasons. Plant height at the harvest of the Sakha 106 rice cultivar increased following the application of the combination of 75% SSP basal application and the foliar spraying of PSBs + PNPs + PA (96.33 and 95.00 cm) in 2019 and 2020, respectively, or 100% SSP, followed by top-dressing with PSBs + AMFs, while low values were recorded for the zero-P fertilizer (88.67 and 89.00 cm) in the two seasons.

**Table 2.** Leaf area index (LAI) in the booting stage, dry matter accumulation (DMA) in the booting stage and plant height at harvest of Sakha 106 rice cultivar were affected by arbuscular mycorrhizal fungi, phosphate-solubilizing bacteria and selected chemical phosphorus fertilizers in the 2019 and 2020 seasons.

| Treatment | LAI | | DMA (g m$^2$) Season | | Plant Height (cm) | |
|---|---|---|---|---|---|---|
| | 2019 | 2020 | 2019 | 2020 | 2019 | 2020 |
| Basal application of 100% SSP (P$_1$) | 3.524 ab | 3.324 ab | 394.7 b | 409.2 ab | 95.67 ab | 94.67 b |
| Basal application of 75% SSP (P$_2$) | 2.120 fg | 1.943 f | 199.7 h | 208.8 g | 89.33 d | 89.33 d |
| P$_2$ + top-dressing with PSBs | 2.677 c | 2.520 c | 223.9 fg | 240.2 f | 90.67 c | 90.00 cd |
| P$_2$ + top-dressing with AMFs | 2.914 bc | 3.000 a–d | 239.4 f | 262.3 e | 90.67 c | 91.00 c |
| P$_2$ + foliar spraying of PNPs | 2.160 f | 2.038 e | 214.7 g | 212.0 g | 89.33 d | 89.67 d |
| P$_2$ + foliar spraying of PA | 2.283 d | 2.117 e | 204.7 gh | 216.5 g | 90.00 cd | 90.00 cd |
| P$_2$ + top-dressing with (PSBs + AMFs) | 3.016 b | 2.787 b | 329.4 c | 352.6 b | 95.67 ab | 94.67 b |
| P$_2$ + foliar spraying of (PSBs + PNPs) | 2.927 bc | 2.473 c | 301.1 d | 315.1 cd | 94.33 b | 95.33 a |
| P$_2$ + foliar spraying of (PSBs + PA) | 2.890 b–f | 2.240 d | 299.2 de | 330.4 c | 94.67 b | 95.67 a |
| P$_2$ + foliar spraying of (PNPs + PA) | 2.510 cd | 2.387 cd | 279.8 e | 301.0 d | 90.67 c | 91.00 c |
| P$_2$ + foliar spraying of (PSBs + PNPs + PA) | 3.706 a | 3.527 a | 464.3 a | 462.8 a | 96.33 a | 95.00 ab |
| Zero P (Control) | 1.960 e | 1.667 g | 179.3 i | 194.6 h | 88.67 d | 89.00 d |
| F. Test | ** | ** | ** | ** | ** | ** |

** = $p < 0.05$. SSP, single super phosphate (15% $P_2O_5$); PSBs, phosphate-solubilizing bacteria; AMFs, arbuscular mycorrhizal fungi; PNPs, phosphorus nanoparticles; PA, phosphoric acid. Different alphabetic letters represent the significant differences among the treatments at $p < 0.05$, according to Duncan's test.

### 3.2. Yield Attributing Characteristics

The results presented in Table 3 revealed that combining 75% SSP basal treatment with the foliar spraying of PSBs + PNPs + PA or 100% SSP considerably enhanced the number of panicles m$^{-2}$ at harvest (506.3 and 521.9) in 2019 and 2020, respectively, while 75%



SSP + the foliar spraying of PSBs + PNPs or 75% SSP + top-dressing with PSBs + AMFs had no differences with respect to 75% SSP + the foliar spraying of PNPs + PA. Low values were recorded for the zero-P fertilizer (271.2 and 311.3) in 2019 and 2020, respectively. The combined application of 75% SSP basal application with the foliar spraying of PSBs + PNPs + PA or 75% SSP + P2 + the foliar spraying of PNPs + PA considerably enhanced the filled grain weight panicle$^{-1}$ (3.549 and 3.627 g) in 2019 and 2020, respectively, compared with the paltry value obtained with the zero-P fertilizer (2.964 and 3.045 g) in 2019 and 2020, respectively. The percentage of filled grains in the Sakha 106 rice cultivar increased with 75% SSP basal application and the foliar spraying of PSBs + PNPs + PA (96.19 and 95.43 %) in 2019 and 2020, respectively, without any differences with respect to 75% SSP + PNPs + PA or the combination of 75% SSP + top-dressing with PSBs + AMFs or 100% SSP, while low values were recorded for the zero-P fertilizer (94.66 and 93.75 %) in 2019 and 2020, respectively.

**Table 3.** Number of panicles, filled grain weight per panicle and filled grain percentage of Sakha 106 rice cultivar were affected by arbuscular mycorrhizal fungi, phosphate-solubilizing bacteria and selected chemical phosphorus fertilizers in the 2019 and 2020 seasons.

| Treatment | No. of Panicles (m$^2$) | | Filled Grain Weight (g Panicle$^{-1}$) Season | | Filled Grains (%) | |
|---|---|---|---|---|---|---|
| | 2019 | 2020 | 2019 | 2020 | 2019 | 2020 |
| Basal application of 100% SSP (P$_1$) | 521.1 a | 547.1 a | 3.534 ab | 3.767 a | 96.47 a | 96.24 a |
| Basal application of 75% SSP (P$_2$) | 312.5 e | 318.6 f | 2.763 de | 2.979 c | 94.92 b | 94.38 b |
| P$_2$ + top-dressing with PSBs | 374.7 d | 398.0 e | 2.997 d | 3.334 b | 94.74 b | 94.09 b |
| P$_2$ + top-dressing with AMFs | 395.7 c | 420.7 d | 3.173 c | 3.249 b | 95.36 ab | 96.01 a |
| P$_2$ + foliar spraying of PNPs | 328.2 e | 322.0 f | 2.763 de | 2.934 c | 94.93 b | 94.44 b |
| P$_2$ + foliar spraying of PA | 366.9 d | 376.8 ef | 3.154 c | 3.239 b | 95.35 ab | 95.25 ab |
| P$_2$ + top-dressing with (PSBs + AMFs) | 445.8 b | 456.5 c | 3.369 bc | 3.546 ab | 96.39 a | 95.89 ab |
| P$_2$ + foliar spraying of (PSBs + PNPs) | 453.8 b | 473.8 b | 3.429 b | 3.405 ab | 95.42 ab | 95.56 ab |
| P$_2$ + foliar spraying of (PSBs + PA) | 444.4 b | 444.5 c | 2.991 d | 3.315 b | 95.61 ab | 96.37 a |
| P$_2$ + foliar spraying of (PNPs + PA) | 416.3 bc | 450.1 c | 3.597 a | 3.739 a | 95.36 ab | 94.96 b |
| P$_2$ + foliar spraying of (PSBs + PNPs + PA) | 506.3 ab | 521.9 ab | 3.549 ab | 3.627 a | 96.19 a | 95.43 ab |
| Zero P (Control) | 271.2 f | 311.3 fg | 2.964 d | 3.045 c | 94.66 b | 93.75 b |
| F. Test | ** | ** | ** | ** | ** | ** |

** = $p < 0.05$. SSP, single super phosphate (15% $P_2O_5$); PSBs, phosphate-solubilizing bacteria; AMFs, arbuscular mycorrhizal fungi; PNPs, phosphorus nanoparticles; PA, phosphoric acid. Different alphabetic letters represent the significant differences among the treatments at $p < 0.05$, according to Duncan's test.

### 3.3. Phosphorus Uptake in Grain and Straw

The data in Figure 2 showed that combining 75% SSP basal treatment with the foliar spraying of PSBs + PNPs + PA significantly enhanced the phosphorus uptake in grain (24.37 and 24.49 kg P ha$^{-1}$) in 2019 and 2020, respectively, without any difference with respect to 100% SSP; furthermore, 75% SSP + the foliar spraying of PSBs + PNPs or 75% SSP + top-dressing with PSBs + AMFs had no differences with respect to 75% SSP + the foliar spraying of PSBs + PA. Low values were recorded for the zero-P fertilizer (17.62 and 17.29 kg P ha$^{-1}$) in 2019 and 2020, respectively. The treatment that combined 75% SSP basal treatment with the foliar spraying of PSBs + PNPs + PA significantly increased phosphorus uptake in straw (17.7 and 17.0 kg P ha$^{-1}$) in 2019 and 2020, respectively, without any difference with respect to 100% SSP; furthermore, 75% SSP + the foliar spraying of PSBs + PNPs or 75% SSP + top-dressing with PSBs + AMFs had no differences compared to 75% SSP + the foliar spraying of PSBs + PA. Low values were recorded for the zero-P fertilizer (10.60 and

10.78 kg P ha$^{-1}$) in 2019 and 2020, respectively, in the Sakha 106 Egyptian rice cultivar (Figure 3).

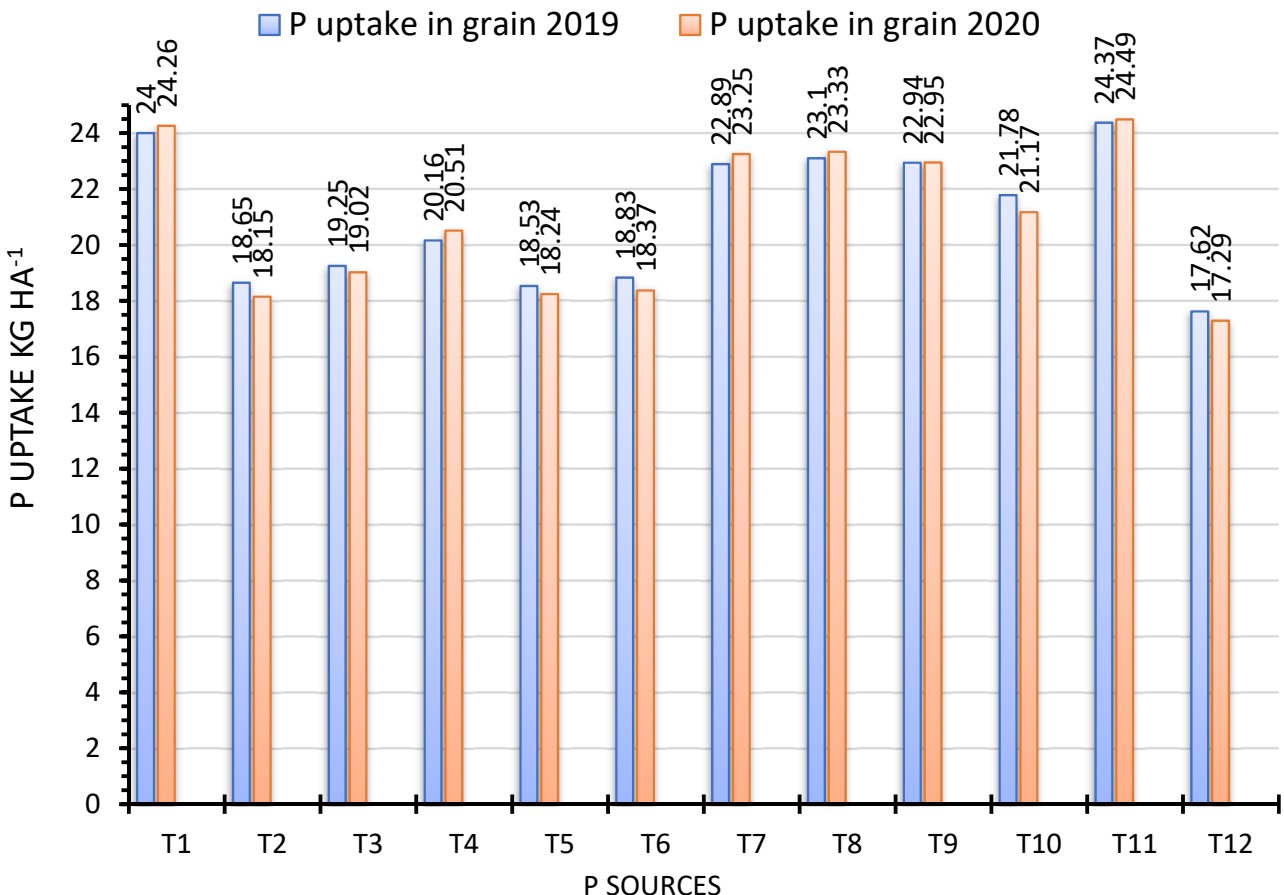

**Figure 2.** Phosphorus uptake in grain of Sakha 106 rice cultivar was affected by arbuscular mycorrhizal fungi, phosphate-solubilizing bacteria and selected chemical phosphorus fertilizers in the 2019 and 2020 seasons. SSP, single super phosphate (15% $P_2O_5$); PSBs, phosphate-solubilizing bacteria; AMFs, arbuscular mycorrhizal fungi; PNPs, phosphorus nanoparticles; PA, phosphoric acid. T1, basal application of 100% single super phosphate (SSP) ($P_1$); T2, basal application of 75% SSP ($P_2$); T3, $P_2$ + top-dressing with phosphate-solubilizing bacteria (PSBs); T4, $P_2$ + top-dressing with arbuscular mycorrhizal fungi (AMFs); T5, $P_2$ + foliar spraying of phosphorus nanoparticles (PNPs); T6, $P_2$ + foliar spraying of phosphoric acid (PA); T7, $P_2$ + top-dressing with PSBs + AMFs; T8, $P_2$ + foliar spraying of (PSBs + PNPs); T9, $P_2$ + foliar spraying of (PSBs + PA); T10, $P_2$ + foliar spraying of (PNPs + PA); T11, $P_2$ + foliar spraying of (PSBs + PNPs + PA); T12, zero-phosphorus fertilizer (Control).

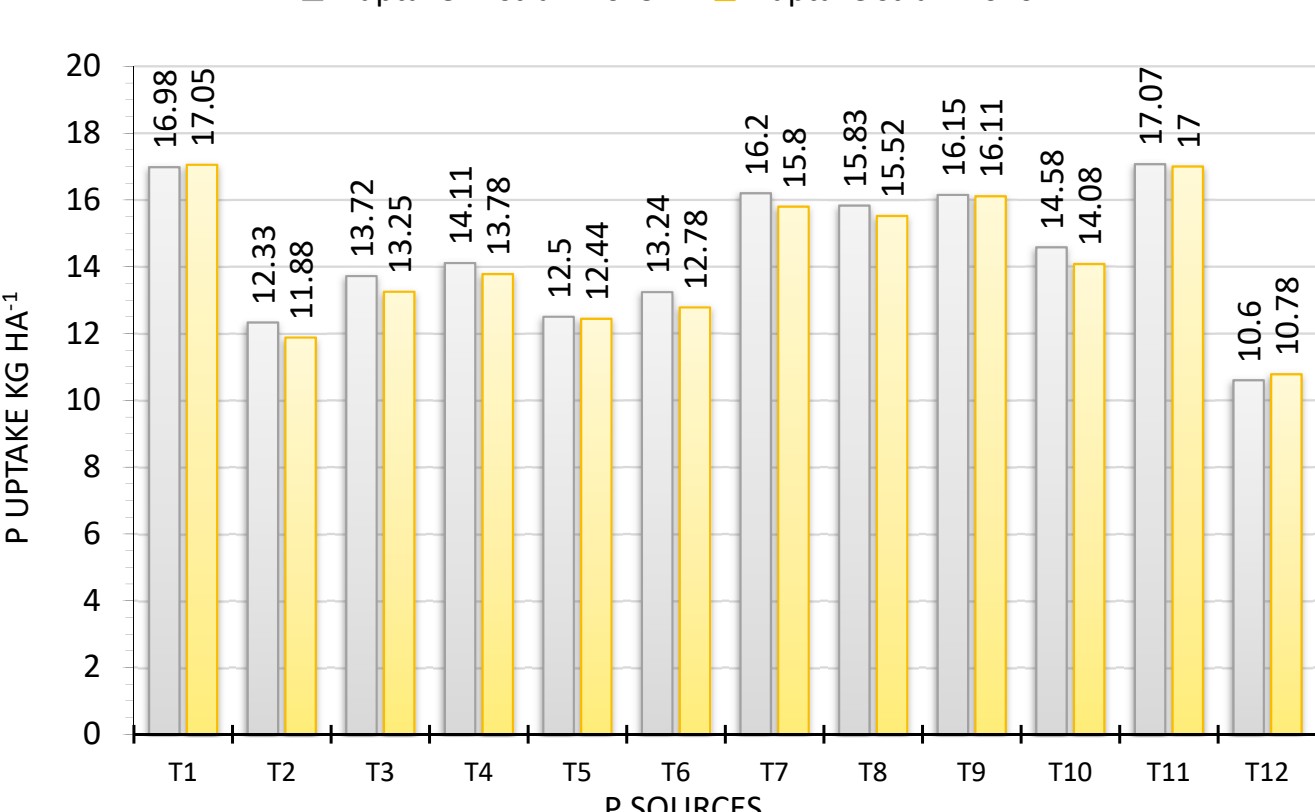

**Figure 3.** Phosphorus uptake in straw of Sakha 106 rice cultivar was affected by arbuscular mycorrhizal fungi, phosphate-solubilizing bacteria and selected chemical phosphorus fertilizers following the application of different P sources in the 2019 and 2020 seasons. SSP, single super phosphate (15% $P_2O_5$); PSBs, phosphate-solubilizing bacteria; AMFs, arbuscular mycorrhizal fungi; PNPs, phosphorus nanoparticles; PA, phosphoric acid. T1, basal application of 100% single super phosphate (SSP) (P1); T2, basal application of 75% SSP (P2); T3, P2 + top-dressing with phosphate-solubilizing bacteria (PSBs); T4, P2 + top-dressing with arbuscular mycorrhizal fungi (AMFs); T5, P2 + foliar spraying of phosphorus nanoparticles (PNPs); T6, P2 + foliar spraying of phosphoric acid (PA); T7, P2 + top-dressing with PSBs + AMFs; T8, P2 + foliar spraying of (PSBs + PNPs); T9, P2 + foliar spraying of (PSBs + PA); T10, P2 + foliar spraying of (PNPs + PA); T11, P2 + foliar spraying of (PSBs + PNPs + PA); T12, zero-phosphorus fertilizer (Control).

*3.4. Available Phosphorus in the Soil after Harvest*

In the 2019 and 2020 seasons, the combination of different phosphorus sources and forms influenced the available phosphorus in the soil after harvest (Figure 4). The availability of phosphorus in the soil after harvest was greatly improved by combining 75% SSP basal treatment with the foliar spraying of PSBs + PNPs + PA (21.75 and 21.70 ppm) in 2019 and 2020, respectively, with no differences with respect to 100% SSP, followed by 75% SSP + top-dressing with PSBs + AMFs, 75% SSP + the foliar spraying of PSBs + PNPs or 75% SSP + the foliar spraying of PSBs + PA. Low availability of P in soil was found when no P was added to the soil (14.77 and 14.39 ppm) in 2019 and 2020, respectively.

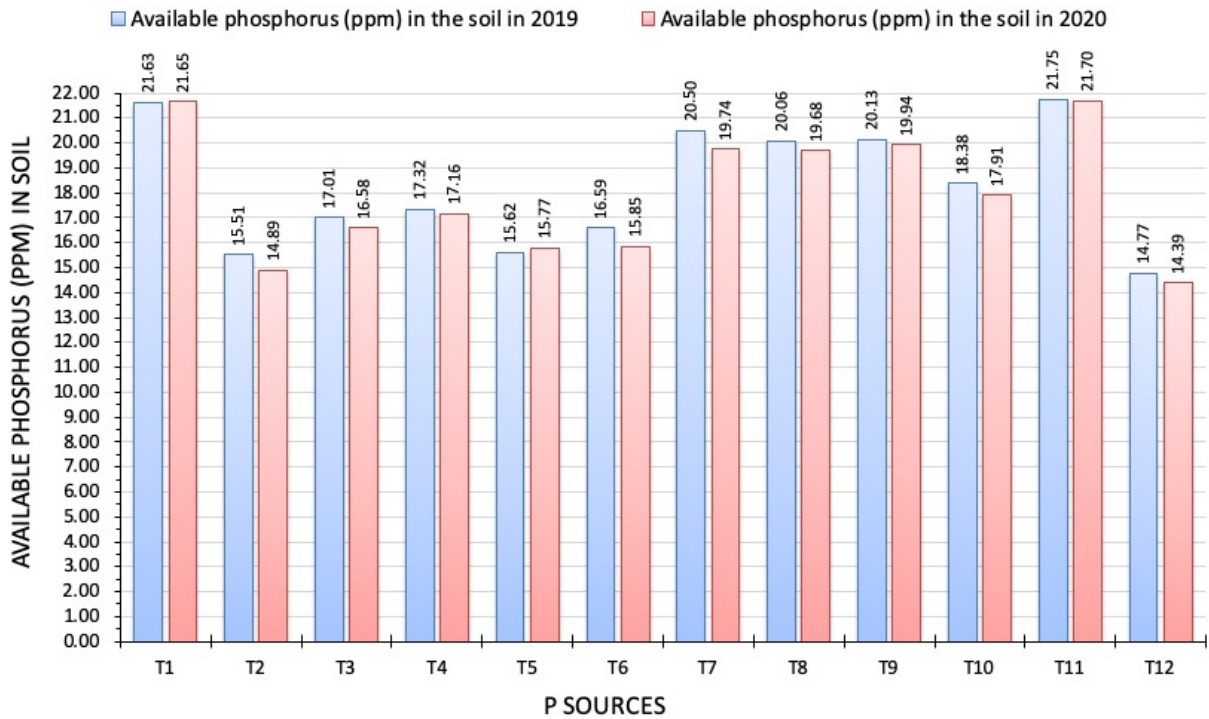

**Figure 4.** Available phosphorus in the soil at harvest was affected by arbuscular mycorrhizal fungi, phosphate-solubilizing bacteria and selected chemical phosphorus fertilizers in Sakha 106 rice cultivar in the 2019 and 2020 seasons. SSP, single super phosphate (15% $P_2O_5$); PSBs, phosphate-solubilizing bacteria; AMFs, arbuscular mycorrhizal fungi; PNPs, phosphorus nanoparticles; PA, phosphoric acid. T1, basal application of 100% single super phosphate (SSP) ($P_1$); T2, basal application of 75% SSP ($P_2$); T3, $P_2$ + top-dressing with phosphate-solubilizing bacteria (PSBs); T4, $P_2$ + top-dressing with arbuscular mycorrhizal fungi (AMFs); T5, $P_2$ + foliar spraying of phosphorus nanoparticles (PNPs); T6, $P_2$ + foliar spraying of phosphoric acid (PA); T7, $P_2$ + top-dressing with PSBs + AMFs; T8, $P_2$ + foliar spraying of (PSBs + PNPs); T9, $P_2$ + foliar spraying of (PSBs + PA); T10, $P_2$ + foliar spraying of (PNPs + PA); T11, $P_2$ + foliar spraying of (PSBs + PNPs + PA); T12, zero-phosphorus fertilizer (Control).

*3.5. Grain and Straw Yields*

According to the data in Figure 5, 100% SSP increased grain yield productivity and produced 9.353 and 9.311 t ha$^{-1}$ in 2019 and 2020, respectively, representing increases of 52.08 and 56.17% compared with the zero-P fertilizer in both seasons, with no differences compared to combining 75% SSP basal application with the foliar spraying of PSBs + PNPs + PA, which increased grain yield productivity to 9.221 and 9.184 t ha$^{-1}$ in 2019 and 2020, respectively, representing increases of 49.93 and 53.44% compared with the zero-P fertilizer in both seasons. In addition, 75% SSP + top-dressing with PSBs + AMFs, 75% SSP + the foliar spraying of PSBs + PNPs or 75% SSP + the foliar spraying of PSBs + PA increased 106 rice grain yield in both seasons, with increases of 45.32 and 50.57%, 47.80 and 52.13% or 35.06 and 40.19 compared with the zero-P fertilizer in both seasons. In contrast, not applying phosphorus fertilizers reduced productivity by 3.203 and 3.349 t ha$^{-1}$ in both seasons, resulting in the low values of 6.150 and 5.962 t ha$^{-1}$ for the zero-P fertilizer in both seasons. Straw yield increased with 100% SSP to 11.510 and 11.820 t ha$^{-1}$, with no differences compared to the 75% SSP + PSBs + PNPs + PA and 75% SSP + PSBs + AMFs treatments, which improved the yield to 11.46 and 11.69 t ha$^{-1}$ in 2019 and 2020, respectively. On the other hand, in 2019 and 2020, the zero-P fertilizer lowered productivity by 3.124 and 3.107 t ha$^{-1}$, respectively, resulting in the low values of 8.386 and 8.713 t ha$^{-1}$ (Figure 6).

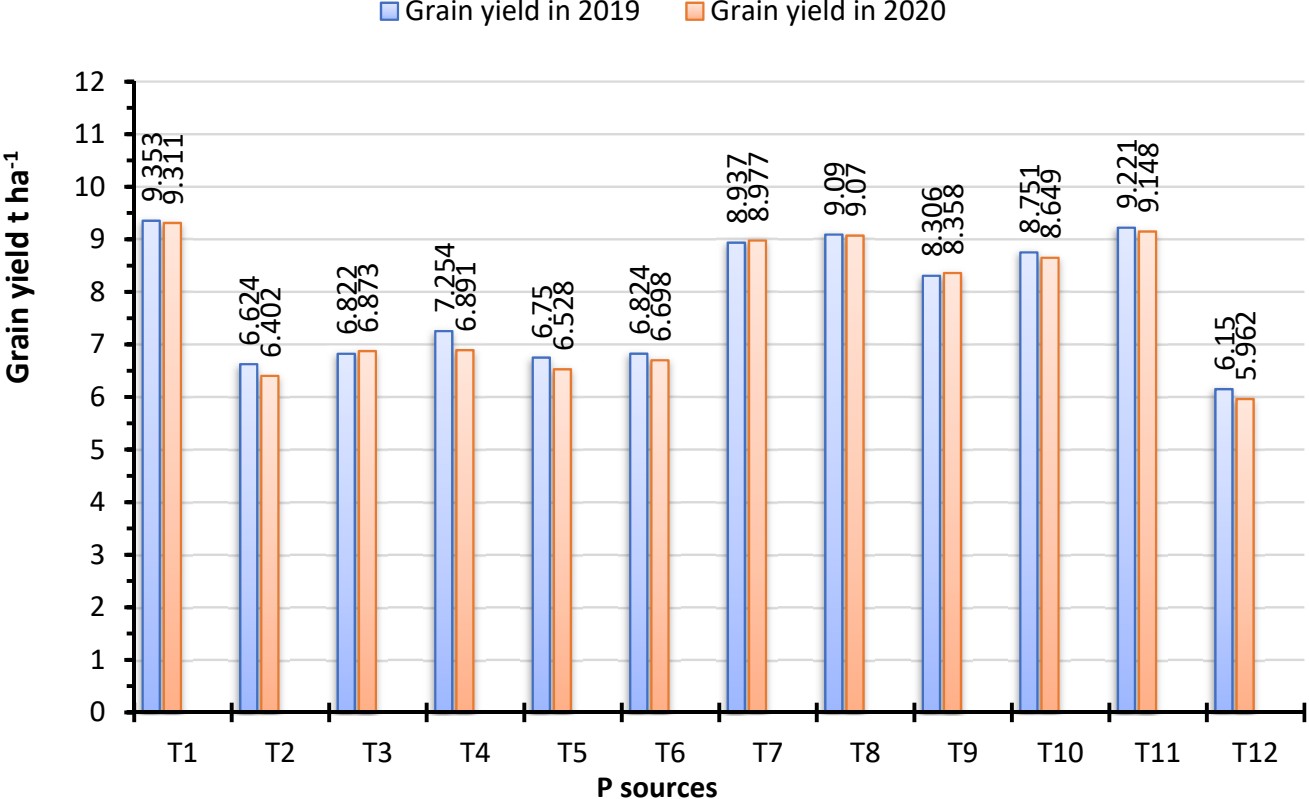

**Figure 5.** Grain yield of Sakha 106 rice cultivar was affected by arbuscular mycorrhizal fungi, phosphate-solubilizing bacteria and selected chemical phosphorus fertilizers in the 2019 and 2020 seasons. SSP, single super phosphate (15% $P_2O_5$); PSBs, phosphate-solubilizing bacteria; AMFs, arbuscular mycorrhizal fungi; PNPs, phosphorus nanoparticles; PA, phosphoric acid. T1, basal application of 100% single super phosphate (SSP) (P1); T2, basal application of 75% SSP (P2); T3, P2 + top-dressing with phosphate-solubilizing bacteria (PSBs); T4, P2 + top-dressing with arbuscular mycorrhizal fungi (AMFs); T5, P2 + foliar spraying of phosphorus nanoparticles (PNPs); T6, P2 + foliar spraying of phosphoric acid (PA); T7, P2 + top-dressing with PSBs + AMFs; T8, P2 + foliar spraying of (PSBs + PNPs); T9, P2 + foliar spraying of (PSBs + PA); T10, P2 + foliar spraying of (PNPs + PA); T11, P2 + foliar spraying of (PSBs + PNPs + PA); T12, zero-phosphorus fertilizer (Control).

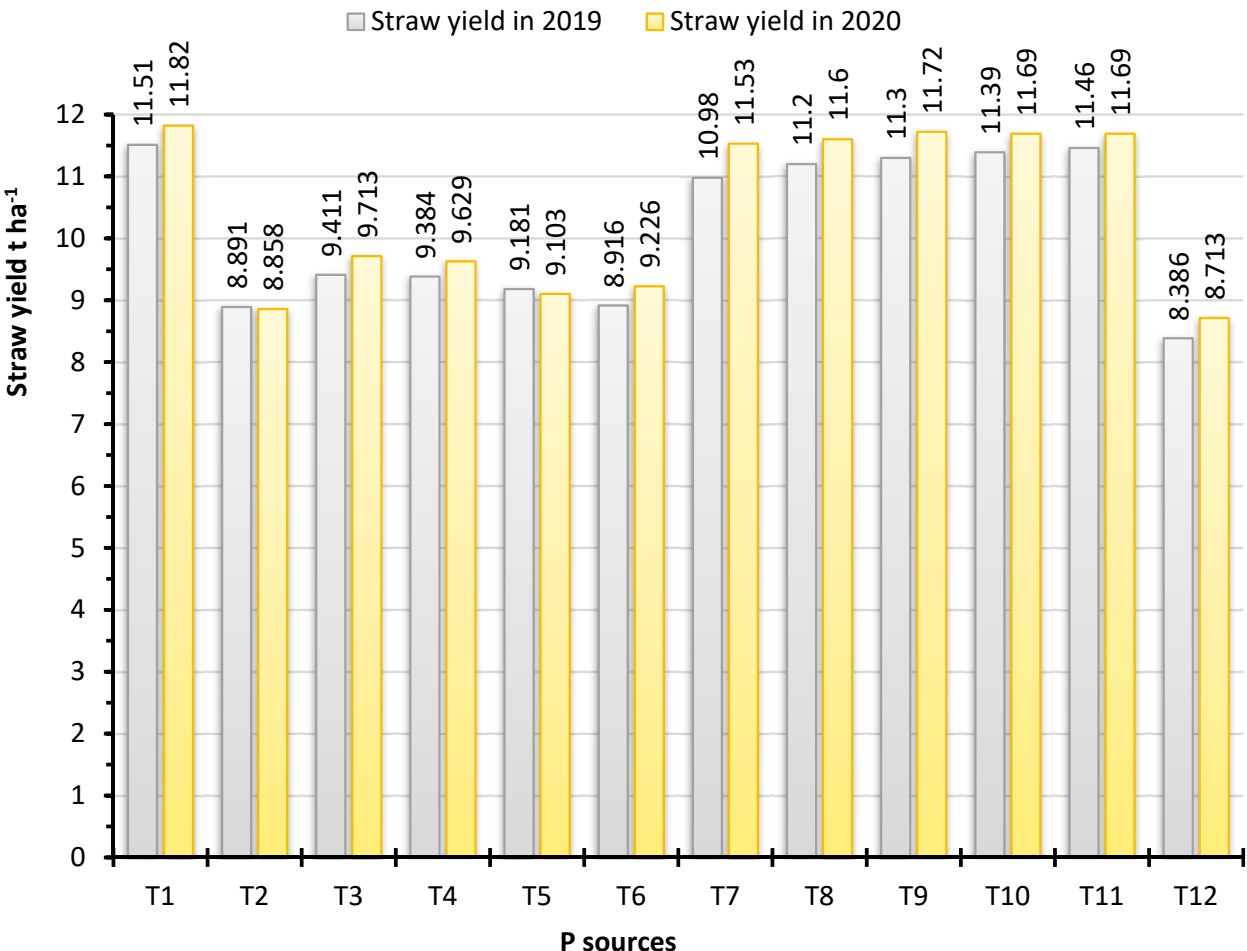

**Figure 6.** Straw yield of Sakha 106 rice cultivar was affected by arbuscular mycorrhizal fungi, phosphate-solubilizing bacteria and selected chemical phosphorus fertilizers in the 2019 and 2020 seasons. SSP, single super phosphate (15% $P_2O_5$); PSBs, phosphate-solubilizing bacteria; AMFs, arbuscular mycorrhizal fungi; PNPs, phosphorus nanoparticles; PA, phosphoric acid. T1, basal application of 100% single super phosphate (SSP) ($P_1$); T2, basal application of 75% SSP ($P_2$); T3, $P_2$ + top-dressing with phosphate-solubilizing bacteria (PSBs); T4, $P_2$ + top-dressing with arbuscular mycorrhizal fungi (AMFs); T5, $P_2$ + foliar spraying of phosphorus nanoparticles (PNPs); T6, $P_2$ + foliar spraying of phosphoric acid (PA); T7, $P_2$ + top-dressing with PSBs + AMFs; T8, $P_2$ + foliar spraying of (PSBs + PNPs); T9, $P_2$ + foliar spraying of (PSBs + PA); T10, $P_2$ + foliar spraying of (PNPs + PA); T11, $P_2$ + foliar spraying of (PSBs + PNPs + PA); T12, zero-phosphorus fertilizer (Control).

## 4. Discussion

The combination of 75% single super phosphate (SSP) basal application with the foliar spraying of phosphate-solubilizing bacteria (PSBs) + phosphorus nanoparticles (PNPs) + phosphoric acid (PA) significantly increased the leaf area index (LAI), dry matter accumulation (DMA) and plant height of the Sakha 106 Egyptian rice cultivar. The combination of chemical and biological techniques as sources of P could favorably boost the efficiency of P resources and optimize the inorganic fertilizer application for crop development, which may explain the increases in the LAI, DMA and plant height [24,51]. Bio- and nanofertilizers have a variety of metabolic effects on plants, increasing photosynthesis rates and increasing rice dry matter and yield. PSB activity, particularly PSB interactions with plants, can help plants to develop by solubilizing P in soil [52]. PSBs promote plant development by generating essential nutrients, such as indole acetic acid and phytohormones [15,53]. The development of rice plants was positively affected by nanoparticles [37,38]. There were no significant differences in combining 75% SSP basal

application with the foliar spraying of PSBs + PNPs + PA, 75% SSP + top-dressing with PSBs + AMFs, 75% SSP + the foliar spraying of PSBs + PNPs or 75% SSP + the foliar spraying of PSBs + PA. Calcium phosphate nanoparticles (CaPO$_4$ NPs) demonstrate synergistic growth stimulation and root reproduction with AMFs and may be developed as a cost-effective nanofertilizer with high effectiveness [54]. Simultaneous inoculation with AMF fungi and PSBs may increase plant profit from insoluble P sources and stimulate plant development better than inoculation with either P microbes or chemical P alone. Adding chemical or biological phosphorus alone may be responsible for low values and may reach zero-P fertilizer values [55,56]. Previous research showed that co-inoculation with AMFs and PSBs might improve plant biomass, P absorption in plants [57,58], soil microbial activity [58] and other soil characteristics [59]. By combining a basal treatment of 75% SSP with the foliar spraying of PSBs+ PNPs+ PA, the number of panicles at harvest, filled grain weight and filled grains percentage of the Sakha 106 rice cultivar considerably increased. The main insoluble P became soluble P via PSB amendment, resulting in an increase in the number of panicles, filled grain weight and filled grains percentage of the Sakha 106 rice cultivar [60]. Root development, enhanced stem strength, higher seed germination, earlier crop maturity, crop quality and increased resistance to plant diseases are some of the growth factors linked to phosphorus [61]. There were no significant differences in combining 75% SSP basal application with the foliar spraying of PSBs + PNPs + PA, 75% SSP + top-dressing with PSBs + AMFs, 75% SSP + the foliar spraying of PSBs + PNPs or 75% SSP + the foliar spraying of PSBs + PA and 100% SSP as the suggested chemical phosphorus dosages. With no fertilizer, the lowest value set a new record. Beneficial microbes are crucial not just for reducing mineral fertilizer use and for being ecologically friendly but also for improving soil quality [62,63]. Bio NPK + 75% inorganic NPK were suggested to improve the number of panicles at harvest, filled grain weight and filled grains percentage, resulting in a high net value (859.1 USD ha$^{-1}$) and decreasing chemical N, P and K by 25%, resulting in a cleaner environment and soil maintenance [13]. Phosphorus uptake in the grain and straw of the Sakha 106 Egyptian rice cultivar was considerably enhanced by combining basal treatment with 75% SSP and the foliar spraying of PSBs + PNPs+ PA. P uptake in grain and straw may have increased due to a higher P content in rice roots, which served as effective bio-inoculants for rice [64,65]. Inoculation with AMFs enhances rice root nutrient uptake by increasing the bioavailability of soil phosphorus for plants in the rhizosphere area and making it easier for rice to absorb phosphorus. Immobile P causes the formation of a phosphate-depleted zone surrounding the roots, although mycorrhizal development helps the roots to rapidly absorb phosphate ions at the root surface. In rice, PSB biofertilizer might improve grain yield by 1–11% and grain P absorption by 6–8% compared with the control, and PSBs fix P biologically to change it from an insoluble form to a soluble and available form [66]. The P concentration in grain was greater than that in straw at harvest. This might have been due to the transfer of P from the shoots to the grain before harvest [67]. AMFs and PSBs have the potential to increase the concentration of P in plant tissues [57,58]. PSB inoculation improves plant P absorption and crop production [23,68]. PSBs may be used in conjunction with inorganic fertilizer dosages to optimize plant development while reducing chemical fertilizer inputs [69,70]. The combination of 75% SSP and the foliar spraying of PSBs, PNPs and PA or of 75% SSP and top-dressing with PSBs and AMFs showed no significant differences when compared to 100% SSP as recommended doses of chemical phosphorus. The lowest value achieved a record with the zero-phosphorus fertilizer. The use of PSB biofertilizers as inoculants improves P availability and absorption by plants because microorganisms generate organic acids and lower soil pH [71,72]. If soils have limited mobile P and high P fixation ability, the amount of P absorbed by plant roots may be too low to meet crop demands [73]. The capacity of a plant to take up P is greatly influenced by its root allocation near P sited in soil [17,74]. The combination of 75% SSP basal treatment and the foliar spraying of PSBs+ PNPs+ PA substantially enhanced the available phosphorus in soil after harvest. The increase in accessible P in paddy soil during harvest might have been attributed to PSBs solubilizing inorganic P to organic acids [73].

The combinations of 75% SSP and PSBs, PNPs and PA, of 75% SSP and top-dressing with PSBs and AMFs, and 75% SSP showed no significant changes when compared to 100% SSP as suggested dosages of chemical P fertilizers. The lowest value achieved a record with the zero-phosphorus fertilizer. Total soil phosphorus is unavailable for absorption due to rapid immobilization by soil organic and inorganic components [23,75,76]. Chung et al. (2005) [77] showed that effective PSBs dissolved poorly soluble P to insoluble P into releasing forms via acidification, chelation and the production of organic acids. Phosphate availability in soil solution can be determined via the changes in pH and organic acids [78]. The usage of these PSBs as bio-inoculants can enhance soil accessible P, reduce inorganic P fertilizer inputs and reduce pollution [79]. Benefits include increased nutrient use efficiency, increased crop production and reduced soil pollution [80,81]. Plants absorb soluble nutrient ions as randomly as they absorb those from fixed conventional fertilizers [82]. The grain and straw yields of the Sakha 106 rice cultivar were considerably improved by combining basal treatment with 75% SSP and the foliar spraying of PSBs + PNPs+ PA. The increases in grain yield and straw yield might have been attributable to PSBs and AMFs sharing the rhizosphere with beneficial microorganisms and possibly playing a keeper function in plant growth. PSBs as a biofertilizer increase plant P absorption and crop production [23,68,83]. PSB inoculation with inorganic P increases the effectiveness of P fertilizer and reduces plant P requirements by around 25% [84]. There were no significant differences among 75% SSP and the foliar spraying of PSBs, PNPs and PA, 75% SSP and top-dressing with PSBs and AMFs, and 100% SSP as prescribed chemical P dosages. With the 0% P fertilizer, the lowest value was recorded. One of the major methods by which plants release P in soil is the excretion of organic acids from roots [85]. Phosphate-solubilizing microorganisms can solubilize immobile soil P, increasing crop production [86]. PSB inoculation combined with plant-growth-promoting rhizobacteria (PGPRs) inoculation might minimize P fertilizer input by 50% without affecting crop output [14,87,88].

## 5. Conclusions

The application of combinations of biofertilizers, i.e., arbuscular mycorrhizal fungi (AMFs), phosphate-solubilizing bacteria (PSBs) and single super phosphate (SSP), beneficially improved phosphorus bioavailability in soil, converting it from an insoluble form to a soluble and available form that plants could absorb, thus enhancing the growth and productivity of rice. When compared with the applied biological or chemical P fertilizer alone, the combination of two biofertilizers (AMFs and PSBs) and one of the chemical phosphorus fertilizers, i.e., single super phosphate (SSP), orthophosphoric acid or Hydroxyapatite, showed the highest crop productivity and improved all examined characteristics. The findings of this study showed that the combination of the foliar spraying of phosphate-solubilizing bacteria (PSBs), top-dressing with arbuscular mycorrhizal fungi (AMFs), the foliar spraying of phosphorus nanoparticles (PNPs), the foliar spraying of phosphoric acid (PA) and the basal application of 75% single super phosphate (SSP) improved the grain yield of the Sakha 106 Egyptian rice cultivar, helping plants and soil by solubilizing fixed P in Egyptian paddy soil and reducing chemical P fertilizers by 25%, thus lowering the use of chemical P fertilizers, reducing P leaching and minimizing pollution. Based on the findings of this study, future research should focus on the effects of using AMFs and PSBs in various forms and application methods instead of chemical P fertilizers, with different irrigation intervals, on soil P content, P utilization efficiency and P uptake in rice.

**Author Contributions:** Conceptualization, N.M.E., M.M.A.A.-A., A.A., K.S.A., R.M.A. and R.A.A.; Data curation, N.M.E., M.M.A.A.-A. and K.S.A.; Formal analysis, N.M.E., M.M.A.A.-A. and K.S.A.; Funding acquisition, A.A., K.S.A., R.M.A. and R.A.A.; Investigation, N.M.E., A.A., K.S.A., R.M.A. and R.A.A.; Methodology, N.M.E. and M.M.A.A.-A.; Project administration, N.M.E., M.M.A.A.-A., A.A., K.S.A., R.M.A. and R.A.A.; Resources, N.M.E. and M.M.A.A.-A.; Software, N.M.E., M.M.A.A.-A., K.S.A. and R.M.A.; Supervision, M.M.A.A.-A.; Validation, N.M.E., M.M.A.A.-A., A.A., K.S.A., R.M.A. and R.A.A.; Visualization, A.A., K.S.A., R.M.A. and R.A.A.; Writing—original draft, N.M.E. and

M.M.A.A.-A.; Writing—review and editing, N.M.E. and M.M.A.A.-A. All authors have read and agreed to the published version of the manuscript.

**Funding:** This research study received no external funding.

**Institutional Review Board Statement:** Not applicable.

**Data Availability Statement:** All data, tables and figures were obtained and produced by the authors of this work.

**Acknowledgments:** The first and scorned authors are grateful to and thank all members of Rice Research Department, Field Crops Research Institute, ARC, Egypt, for the support provided for conducting this research study. Co-author Khalid S. Alshallash would like to thank the deanship of scientific research at Imam Mohammed Bin Saud Islamic University, Saudi Arabia, for supporting the publication of this research work.

**Conflicts of Interest:** The authors declare that they have no conflict of interest.

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
