# Peer review of "Impact of Arbuscular Mycorrhizal Fungi, Phosphate Solubilizing Bacteria and Selected Chemical Phosphorus Fertilizers on Growth and Productivity of Rice"

_agriculture, doi:10.3390/agriculture12101596_

Round 1

Reviewer 1 Report

Title

The title is not clear.

Abstract

- Underscore the scientific value-added to your paper in your abstract. Your abstract should clearly state the essence of the problem you are addressing, what you did and what you found and recommend. That will help a prospective reader of the abstract to decide if they wish to read the entire article.

- The experimental treatments should be added in this section.

- Lines 22-26: Which of the fertilizers used was the best?

- Lines 22-26: Please add the values of measured traits. also, add the increasing percentage.

- What is the best recommendation for rice producers?

Keywords

- Don’t use abbreviation words in Keywords.

Introduction

- Line 37: ‘ton ha-1’ should be modified.

- The introduction section is too short.

- The linkage between paragraphs is missed.

- Lines 41: It has a limited role?

- Add more information about the stabilization of phosphorus and the lack of access to P in the soil by the roots of plants.

- Please subject the manuscript to review made by English Native speaker.

- Justify novelty in Introduction and Discussion.

- Please add the objective of study at the end of introduction.

- The hypothesis should be added in introduction?

- The novelty is missed. In introduction, the authors only introduce the different fertilizer types for nutrient.

Materials and methods

this section should be revised deeply.

- The treatments should be added in this section. It’s not clear. Do you use phosphate solubilizing bacteria, arbuscular mycorrhizal fungi separately and integrative with each other? The authors in abstract section noted that ‘chemical P fertilizers combined with foliar spray of PSB, PNPs, and PA….’

- Have you determined the optimal and toxic concentration of nano fertilizer for wheat plant?

- Lines 128-131: Please add the AMF species that used in this study.

- Do you perform combined analysis?

- Do you consider the growing year as separate factor in analysis?

- The combined analysis done with MSTAT-C?

Results

- Please add the highest and lowest values.

- Please the increasing or decreasing percentage

- Lines 191-202: Also add the increasing or decreasing percentage in this section.

- The results section is read difficulty. Please revise all sections of results.

Figure 2: Why the results of 2019 and 2020 were separated to each other?

Figure 2: I recommend to separating the P uptake in straw and grain.

Figure 4: Please separate the grain yield data wish straw yield data.

Discussion

Line 325: According to ????

- Line 325-328: The sentence is not clear.

- Some sentences have been repeated several times in different parts of the discussion. Please revise these.

- Please add the results of harvest index for different treatments.

- The discussion is repetitive. For example, the sentence ‘In rice, PSB biofertilizer might improve grain yield by 1-11% and grain P absorption by 6-8% above control’ is added in the lines of 354-355 and 390-391.

- Lines 393-394: The sentence is incomplete.

- The novelty is missed in the study. The authors should be explained that why integrative application of fertilizers was better than separated application.

Conclusion

- This section is repetitive and should be rewritten.

- Please make sure your conclusions' section underscores the scientific value-added of your paper, and/or the applicability of your findings/results. Highlight the novelty of your study.

- What suggestions do you have for future research in this field?

Reviewer 2 Report

The current manuscript entitled “Effect of Arbuscular Mycorrhizal Fungi, Phosphate Solubilizing Bacteria and Different Sources of Phosphorus on Rice” by Elekhtyar et al. deals with experiments on rice crops under different biofertilizers and phosphorus applications. The study suggested that the combined use of biofertilizers can reduce the net phosphorus fertilizer input and cost for better rice cropping. After a careful reading, I found this work interesting and suitable for publication in Agriculture MDPI. However, I have pointed out some grey points in the manuscript which need to be addressed before the final consideration of this paper. I have noted several typing, grammatical and syntax errors. I suggest a major revision. My specific comments are:

1.      English should be corrected by a native speaker.

2.      Title: Correct it to “Combined Use of Two Biofertilizers (Arbuscular Mycorrhizal Fungi and Phosphate Solubilizing Bacteria) and Selected Phosphorus Fertilizers on Growth and Productivity of Rice.”

3.      In the abstract first sentence, delete “After nitrogen” and rewrite the sentence to take the reader’s attention toward the actual research problem.

4.      The abstract should follow this pattern: Research problem and needs, research objectives, experimental design, major results (numerical), overall outcome, conclusion, and research importance.

5.      Poor keyword selection.

6.      How this work is going to help in minimizing environmental pollution? The authors have mentioned this in the abstract.

7.      Subscript and superscripts need to be corrected in the entire manuscript.

8.      The introduction lacks proper linking of the problem, hypothesis, and research objectives. Too short paragraphs are randomly placed and illogical flow of reading.

9.      Several abbreviations are not defined at their first use.

10.   Provide geocoordinates of experimental sites. Also, provide the exact source of material collection (Make, Model, City, Country, etc.).

11.   Provide brief methodology for soil analysis.

12.   Use uniform SI units (ppm or mgkg-1). Many parameters are not described properly.

13.   TEM analysis instrument information is missing. Poor figure caption writing.

14.   Overall experimental design should be provided as an outline/flow diagram.

15.   What about the treatment replications?

16.   Reference for models and equations are missing.

17.   Provide software details (version, make, city, country).

18.   The discussion needs major improvement in terms of real interaction between plant-microbe and also soil.

Round 2

Reviewer 1 Report

In the revised version, the authors have appropriately edited and revised this earlier version according to the comments and suggestions from the reviewers. However, some comments were not modified in the text. The authors should modify some corrections as follows:

- In the abstract the authors should be added the highest values not units.

- Add more information about the stabilization of phosphorus and the lack of access to P in the soil by the roots of plants.

- The hypothesis should be added in introduction.

- I see some grammatically errors in different sections of manuscript especially in the lines of 252-259. Please check the manuscript based on the grammatically structure.

 - Lines 856-860: The sentence is not clear.

- ‘With percentag 52.08 and 56.17 compared with zero P fertilizer in both seasons’. This sentence was not clear. Increasing or decreasing. Please indicate the mentioned percentage has increased or decreased? Please check all sentences in results and revised these.

- in the first line of conclusion ‘etweeen’ should be changed to ‘between’

- in the figure 6 ‘p source’ should be changed to ‘P sources’.

Author Response

Responses to Review Report (Reviewer 1)  (Round 2)

Comments and Suggestions for Authors

In the revised version, the authors have appropriately edited and revised this earlier version according to the comments and suggestions from the reviewers. However, some comments were not modified in the text. The authors should modify some corrections as follows:

- In the abstract the authors should be added the highest values not units. Done

The results showed that the highest values mostly obtained from the combination of 75% SSP) as a basal application with foliar spray of PSB), PNPs) and PA) that had a substantial beneficial impacts on leaf area index (3.706 & 3.527), dry matter accumulation (464.3 & 462.8 g m2), plant height (96.33 & 95.00 cm), phosphorus uptake in grain (24.3 & 24.49 Kg ha-1), phosphorus uptake in straw (17.7 & 17.0 Kg ha-1), available phosphorus in the soil at harvest (21.75 & 21.70 ppm) in 2019 & 2020 seasons respectively, moreover, 75% SSP as a basal application with foliar spray of PSB, PNPs and PA or 100% SSP as a basal application alone improved the number of panicles (506.3 or 521.1 & 521.9 or 547.1 m-2), filled grain weight (3.549 or 3.534 & 3.627 or 3.767 g panicle-1), , percentage of filled grain(96.19 or 96.47 & 95.43 or 96.24%), grain yield (9.353 or 9.221 & 9.311 or 9.148 t ha-1) and straw yield (11.51 or 11.46 & 11.82 or 11.69 t ha-1) in 2019 & 2020 seasons respectively. Chemical P fertilizers combined with foliar spray of PSB, PNPs, and PA produced the highest crop productivity and improved the most of the examined characteristics without any significant changes with chemical P application alone on some other characteristics, then 75% SSP + top-dressed of (PSB + AMF)

- Add more information about the stabilization of phosphorus and the lack of access to P in the soil by the roots of plants. Done

Phosphorus fixing in Egyptian soil is a big problem in agricultural production. P can't move far in the soil to go to the roots [13]. Salts of phytic acid are the main form of organic P in the soil and aren't available to plants and it is fixed in soil colloids in Ca, Al and Fe phosphates form, [14]. Inorganic P fertilizer efficiency is very low at 10 to 20% only [15], Phosphate is only absorbed by plants as monobasic (H2PO4)- and dibasic (HPO4)-2 ions. A large amount of soluble inorganic phosphate applied to soil as inorganic fertilizer is rapidly mobilized and unavailable to plants shortly after application. Thus, increasing soil phosphorus availability requires the release of insoluble and fixed forms of phosphorus. [16]. Some soil microorganisms can mineralize and solubilize P from organic and inorganic form. Soil microorganisms, primarily those of the genera Phosphate-solubilizing bacteria and arbuscular mycorrhizal fungi, can convert insoluble phosphates to soluble forms by secreting organic acids [20]. On the other hand, conventional farming relies heavily on the application of chemical phosphorous fertilizer to maintain optimal levels of phosphorous in agricultural soils. The majority of phosphates in the soil are absorbed by soil particles or incorporated into soil organic matter [91]. A large portion of the soluble phosphate applied to the soil chemical fertilizer is quickly immobilized and plants are rendered inaccessible. Thus, P is lost by leaching losses, causing environmental pollution problems. Current searches are attempting to overcome this condition by investigating alternative sources that are both cost effective and environmentally friendly [19].

- The hypothesis should be added in introduction. Done

- I see some grammatically errors in different sections of manuscript especially in the lines of 252-259. Please check the manuscript based on the grammatically structure. Corrected

 - Lines 856-860: The sentence is not clear. Done

- ‘With percentage 52.08 and 56.17 compared with zero P fertilizer in both seasons’. This sentence was not clear. Increasing or decreasing. Please indicate the mentioned percentage has increased or decreased? Please check all sentences in results and revised these. Done

- in the first line of conclusion ‘etweeen’ should be changed to ‘between’ Done

- in the figure 6 ‘p source’ should be changed to ‘P sources’. Done

Reviewer 2 Report

The authors have revised manuscript according to my comments. I suggest acceptance in current form. Thank you.

Author Response

Responses to Review Report (Reviewer 2)  (Round 2)

Dear Prof.

Thank very much you for your efforts, comments and Suggestions.
